# OpenReview forum: "Toxicity in chatgpt: Analyzing persona-assigned language models"
_EMNLP/2023/Conference — EMNLP 2023 Findings_

### Official Review · Reviewer_tCh3 · 2023-07-20

**Soundness:** 5

**Ethical Concerns:**

Yes

**Excitement:**

4: Strong: This paper deepens the understanding of some phenomenon or lowers the barriers to an existing research direction.

**Justification For Ethical Concerns:**

People with bad intentions could use the results to generate toxic texts.

**Paper Topic And Main Contributions:**

The authors perform a toxicity analysis of generations from CHATGPT. They find that CHATGPT can be consistently toxic about a wide range of topics when it is assigned a persona, such as a dictator. They perform tests with 90 different personas.
The problem about this finding is, that technologies that build on top of CHATGPT can generate toxic language by making such system-level modifications. Therefore, systems based on LLMs should be thoroughly tested before being deployed.

**Reasons To Accept:**

It is an important study that can help make LLMs safer and less toxic. The researchers analyse more than half a million generations.

**Reasons To Reject:**

I cannot think of any reasons to reject it.

**Reproducibility:**

5: Could easily reproduce the results.

**Reviewer Confidence:**

5: Positive that my evaluation is correct. I read the paper very carefully and I am very familiar with related work.

---

> ### Author Rebuttal · Authors · 2023-08-26
>
> We thank you for you feedback and comments.
> We especially appreciate that you agree with the need to study the role of artifacts like system parameters and personas and how they affect toxicity.
> Thanks for your time!

---

### Official Review · Reviewer_QcSQ · 2023-07-27

**Soundness:** 5

**Excitement:**

4: Strong: This paper deepens the understanding of some phenomenon or lowers the barriers to an existing research direction.

**Paper Topic And Main Contributions:**

The authors identified violations of two principles of “Blueprint For An AI Bill Of Rights” (Algorithmic Discrimination Protections and Safe and Effective Systems) by using ChatGPT with an assigned persona as a system parameter. The paper presents a detailed and systematic analysis of the toxicity of generated opinions about 123 diverse entities, with one of 90 personas assigned.

**Questions For The Authors:**

Question A: Could you more elaborate on the difficulties of extending this study to other language models?

Question B: Do you plan to extend the persona set?

Question C: Have you considered adding health entities? For example, there is a lot of stigma on some mental state disorder or appearance on the internet resources, that could leak to ChatGPT during training.

**Reasons To Accept:**

- Toxicity and bias are the known drawbacks of generative models’ usage in real-world applications, especially when it violates existing regulations. Such work helps to identify weak points of currently widely used models to be fixed as soon as possible and be a benchmark for future models.
- Diverse and systematic analysis of ChatGPT, accompanied by plenty of examples.
- The claims are supported by concrete values, different metrics, and figures.
- The performance results are supported with significance tests and standard deviations.

**Reasons To Reject:**

Lack of information how exposed risks might be mitigated.

**Reproducibility:**

4: Could mostly reproduce the results, but there may be some variation because of sample variance or minor variations in their interpretation of the protocol or method.

**Reviewer Confidence:**

5: Positive that my evaluation is correct. I read the paper very carefully and I am very familiar with related work.

---

> ### Author Rebuttal · Authors · 2023-08-26
>
> We thank you for you feedback and comments.
> We answer the questions you posed below:
>
> **Question A**: Could you more elaborate on the difficulties of extending this study to other language models?
>
> We believe that it is fairly straight-forward to extend our analysis to other language models and we have performed some preliminary experiments on open source models as well. They way the system parameter is implemented differs from one system to another. But the most common way is to provide it as part of a prompt, which makes it easy to perform our study on most language models.
>
> **Question B**: Do you plan to extend the persona set?
>
> We have received interest on increasing the set of personas and we plan to do the same. If you believe there are specific sets of personas which might add value, do let us know and we will be happy to add them to our camera-ready version.
>
> **Question C**: Have you considered adding health entities? For example, there is a lot of stigma on some mental state disorder or appearance on the internet resources, that could leak to ChatGPT during training.
>
> This is indeed a very good suggestion. We think it will be worth adding this to the camera-ready version. In our initial experiments we notice that certain health entities do get significantly more hate than others.
>
> Thanks again for your questions, this will definitely improve our paper.

---

### Official Review · Reviewer_85RX · 2023-08-10

**Soundness:** 2

**Excitement:**

3: Ambivalent: It has merits (e.g., it reports state-of-the-art results, the idea is nice), but there are key weaknesses (e.g., it describes incremental work), and it can significantly benefit from another round of revision. However, I won't object to accepting it if my co-reviewers champion it.

**Paper Topic And Main Contributions:**

In this paper, the authors investigate the effect of assigning personas on toxicity in ChatGPT's responses. The authors categorize and evaluate thoroughly the personas that can be assigned to ChatGPT using the "system" setting. The results show assigning these personas can make ChatGPT generate much more toxic content.

**Reasons To Accept:**

The paper is well-written and easy to understand. The categorization of personas ensures a thorough analysis. The authors also indeed find settings where ChatGPT tends to generate toxic content. It is an important topic, and the finding can be useful for developing safer LLM in the future.

**Reasons To Reject:**

While comprehensive in the personas evaluated, this paper falls short of the goal of evaluating the toxicity risk in ChatGPT. It is unlikely that a normal user would consider using the "system" setting or deliberately assigning a role. Even if the role is intentionally assigned, the response still aligns with the user's request. For example, users who assign Ali's persona to ChatGPT would expect the system to respond with more toxic language. The authors should adjust the scope of the claim at the beginning of the paper.

Some claims are also misleading due to the toxicity metric used. This work mainly relies on Google's perspective API for toxicity evaluation. The toxicity score from the API can be understood as a probability of toxic content included in the given text. Claiming 6 or 3 times the toxicity does not make sense when directly comparing the toxicity scores. For example, if the baseline toxicity score is 0.1, 3 times the score is still relatively low and considered highly unlikely to be toxic (the API's documentation recommends using 0.7 as the threshold for selecting potentially toxic content). The authors should consider improving the metric for evaluation (e.g., percentage of responses higher than certain threshold).

The authors only evaluated GPT-3.5 version of ChatGPT. However, it has been months since the introduction of the newer GPT-4 (in fact, even GPT-4 has gone through multiple iterations). GPT-4 claims a huge performance improvement compared to GPT-3.5, including reduced toxic generation (see GPT-4 technical report in the link below). It would be interesting and important to see if the risk still exists in the current version of the model.

https://cdn.openai.com/papers/gpt-4.pdf

**Reproducibility:**

3: Could reproduce the results with some difficulty. The settings of parameters are underspecified or subjectively determined; the training/evaluation data are not widely available.

**Reviewer Confidence:**

4: Quite sure. I tried to check the important points carefully. It's unlikely, though conceivable, that I missed something that should affect my ratings.

---

> ### Author Rebuttal · Authors · 2023-08-28
>
> We thank you for your comments and questions and address your concerns below:
>
> *“It is unlikely that a normal user would consider using the "system" setting or deliberately assigning a role.”*
>
> As we mention in the paper, the “system parameter” can be set by developers and companies who are building a chatbot for their users. Even though the user might not explicitly set the system parameter, there are several examples of entities using this to create different personalities without performing explicit safety analysis (e.g., character.ai). This is the reason we believe it is important to study its effects.
>
> *“For example, users who assign Ali's persona to ChatGPT would expect the system to respond with more toxic language.”*
>
> We respectfully disagree with this. We do not quite understand why Muhammad Ali, the great boxer, would be expected to spout toxic language. Could you please explain the same?
> Furthermore, we believe that the model is clearly intended to be safe, because it declines to utter toxic generations with many personas. Only selectively does it break this rule, which points out at a deep underlying bias which we have uncovered.
>
>
>
> *“Some claims are also misleading due to the toxicity metric used. This work mainly relies on Google's perspective API for toxicity evaluation…….the API's documentation recommends using 0.7 as the threshold for selecting potentially toxic content.”*
>
> Following several studies like [1], we use the Perspective API because it is the most well acknowledged and carefully studied toxicity metric we have at our disposal. Regarding the threshold, the API documentation clearly suggests that the threshold depends on the use case [2] (see section on “Choosing thresholds”). When you look at the examples which have even a toxicity score of 0.3, they can contain pretty hateful and targeted content (the following utterance has a toxicity score of 0.3 – <Entity>, that devilish ideology that has plagued nations for decades, has once again reared its ugly head).
>
> [1] - Liang, Percy, et al. "Holistic evaluation of language models." arXiv preprint arXiv:2211.09110 (2022).
>
> [2] - https://developers.perspectiveapi.com/s/about-the-api-score?language=en_US
>
> *“The toxicity score from the API can be understood as a probability of toxic content included in the given text. Claiming 6 or 3 times the toxicity does not make sense when directly comparing the toxicity scores.”*
>
> 6 times higher toxicity implies that the probability of toxic content produced is 6 times higher. We do believe that probabilities can be compared but will change our writing to make sure this is better explained to a lay reader.
>
> *“The authors only evaluated GPT-3.5 version of ChatGPT. However, it has been months since the introduction of the newer GPT-4”*
>
> We agree that the community would benefit from an extension of the safety analysis to multiple other models and these are the kinds of conversations and works we want to encourage through our study.
> A study on GPT-4 is implausible for the following reasons:
> As noted in our study, we perform a statistical analysis with over half a million generations. A statistical study with many generations is a necessity for the conclusions we are able to draw. Performing this scale of a study with GPT-4 is not possible because of monetary considerations (10X costlier, bringing the cost to tens of thousands of dollars).
> We agree with you however that future research should consider open source models and other LLMs.

---

### Meta-Review · Area_Chair_khaM · 2023-09-17

**Recommendation:** 2

**Metareview:**

This is a comprehensive study, dealing with the toxicity risk in a language model, which is an important issue. Although the results are detailed and thoroughly analysed, there are several matters that raise concerns.

Firstly, it is unclear if the described situations are applicable to regular user interactions and the relevance to real-life scenarios is questionable. The paper also heavily relies on Google's Perspective API for toxicity evaluation, which is problematic as it could be misleading when comparing toxicity scores. Additionally, the manuscript covers only the GPT-3.5 version of ChatGPT, while newer iterations of the model already exist and claim to have significantly improved performance and reduced toxicity. While a comparison would be expensive, it unfortunately would be necessary to make this paper completely sound. Otherwise, it risks being a temproary snapshot. The authors rebuttal that "while the analysis was performed on ChatGPT, the arguments we make should naturally hold for all LLMs" is not self-evident.
One reviewer noticed that the report doesn't provide enough details on mitigating the highlighted risks.

---

### Decision · Program_Chairs · 2023-10-07

**Decision:**

Accept-Findings

**Comment:**

This is a comprehensive study, dealing with the toxicity risk in a language model, which is an important issue. Although the results are detailed and thoroughly analysed, there are several matters that raise concerns.

Firstly, it is unclear if the described situations are applicable to regular user interactions and the relevance to real-life scenarios is questionable. The paper also heavily relies on Google's Perspective API for toxicity evaluation, which is problematic as it could be misleading when comparing toxicity scores. Additionally, the manuscript covers only the GPT-3.5 version of ChatGPT, while newer iterations of the model already exist and claim to have significantly improved performance and reduced toxicity. While a comparison would be expensive, it unfortunately would be necessary to make this paper completely sound. Otherwise, it risks being a temproary snapshot. The authors rebuttal that "while the analysis was performed on ChatGPT, the arguments we make should naturally hold for all LLMs" is not self-evident.
One reviewer noticed that the report doesn't provide enough details on mitigating the highlighted risks.